# Disruption of Redox Homeostasis by Alterations in Nitric Oxide Synthase Activity and Tetrahydrobiopterin along with Melanoma Progression

**DOI:** 10.3390/ijms23115979

**Published:** 2022-05-26

**Authors:** Jaqueline Pereira Moura Soares, Diego Assis Gonçalves, Ricardo Xisto de Sousa, Margareth Gori Mouro, Elisa M. S. Higa, Letícia Paulino Sperandio, Carolina Moraes Vitoriano, Elisa Bachir Santa Rosa, Fernanda Oliveira dos Santos, Gustavo Nery de Queiroz, Roberta Sessa Stilhano Yamaguchi, Gustavo Pereira, Marcelo Yudi Icimoto, Fabiana Henriques Machado de Melo

**Affiliations:** 1Department of Physiological Sciences, Santa Casa de São Paulo School of Medical Sciences, São Paulo 01224-001, Brazil; jpms1995@outlook.com (J.P.M.S.); xisto1987@hotmail.com (R.X.d.S.); robertasessa@gmail.com (R.S.S.Y.); 2Department of Parasitology, Microbiology and Immunology, Juiz de Fora Federal University, Juiz de Fora 36036-900, Brazil; diegoassisg@gmail.com; 3Micro-Imuno-Parasitology Department, Federal University of Sao Paulo, São Paulo 05508-090, Brazil; 4Nefrology Discipline, Federal University of Sao Paulo, São Paulo 05508-090, Brazil; margaret_gori@hotmail.com (M.G.M.); emshiga@gmail.com (E.M.S.H.); 5Department of Pharmacology, Federal University of Sao Paulo, São Paulo 05508-090, Brazil; leticiasperandio7@gmail.com (L.P.S.); gustavo.pereira@unifesp.br (G.P.); 6Department of Pharmacology, Institute of Biomedical Science, Universidade de São Paulo, São Paulo 05505-000, Brazil; car.vitoriano@gmail.com (C.M.V.); elisa.bsr@hotmail.com (E.B.S.R.); fernanda.sts00@outlook.com (F.O.d.S.); guhnery0@gmail.com (G.N.d.Q.); 7Biophysics Department, Federal University of Sao Paulo, São Paulo 05508-090, Brazil; icimoto@unifesp.br; 8Institute of Medical Assistance to Public Servants of the State (IAMSPE), São Paulo 04039-000, Brazil

**Keywords:** melanoma, tetrahydrobiopterin, redox homeostasis, nitric oxide synthase, nitric oxide, reactive oxygen species

## Abstract

Cutaneous melanoma emerges from the malignant transformation of melanocytes and is the most aggressive type of skin cancer. The progression can occur in different stages: radial growth phase (RGP), vertical growth phase (VGP), and metastasis. Reactive oxygen species contribute to all phases of melanomagenesis through the modulation of oncogenic signaling pathways. Tetrahydrobiopterin (BH4) is an important cofactor for NOS coupling, and an uncoupled enzyme is a source of superoxide anion (O_2_^•−^) rather than nitric oxide (NO), altering the redox homeostasis and contributing to melanoma progression. In the present work, we showed that the BH4 amount varies between different cell lines corresponding to distinct stages of melanoma progression; however, they all presented higher O_2_^•−^ levels and lower NO levels compared to melanocytes. Our results showed increased NOS expression in melanoma cells, contributing to NOS uncoupling. BH4 supplementation of RGP cells, and the DAHP treatment of metastatic melanoma cells reduced cell growth. Finally, Western blot analysis indicated that both treatments act on the PI3K/AKT and MAPK pathways of these melanoma cells in different ways. Disruption of cellular redox homeostasis by the altered BH4 concentration can be explored as a therapeutic strategy according to the stage of melanoma.

## 1. Introduction

Cutaneous melanoma is the most aggressive type of skin cancer due to its high heterogeneity and resistance capability. The incidence of melanoma has been rising dramatically over past the few years, which has been associated with increased life expectancy and ultraviolet light (UV) exposure [1,2]. Melanomas represent only 1% of all skin cancers; however, they are responsible for the vast majority of the deaths related to skin cancer [3]. Melanomas emerge from the malignant transformation of melanocytes, melanin-producing cells that are primarily located in the basal layer of the epidermis. Initially, the growth occurs horizontally along with the epidermis, known as the radial growth phase (RGP). The cells acquire invasive capabilities, where, in addition to horizontal dissemination, the tumor invades deeper layers of the skin, a stage known as the vertical growth phase (VGP), increasing the metastatic potential [4]. It is important to mention that development does not always occur in this linear way. Although melanomas can be cured in the early stages through surgical excision, metastatic melanoma is still refractory to available treatments such as immune checkpoint inhibitors and BRAF V600E target therapy [5,6]. According to the American Joint Committee on Cancer (AJCC), melanoma development is classified into five different stages, from 0 (melanoma in situ) to IV (metastatic melanoma). The criteria used by AJCC to characterize melanoma progression is based on the tumor thickness with or without ulceration, the presence of melanoma cells in the lymph nodes and the metastatic niches [7]. Molecular characterization of the complex melanoma landscape confirmed the heterogeneity and high mutational burden of cutaneous melanoma, leading to the classification into four subgroups: mutant *BRAF*, mutant *NRAS*, mutant *NF1*, and triple wild-type (WT) [8]. Identification of the most prevalent spectrum of mutated genes also revealed previously described melanoma oncogenes (*KIT*, *TERT*, and *MITF*) and tumor suppressors, including *CDKN2A*, *TP53*, *PTEN*, and *RB1*, as well as recently identified driver mutations, such as *RAC1*, *MAP2K1*, *PPP6C*, *SNX31*, *ARID2*, and *STK19* [9,10,11]. The pattern of mutated genes showed genomic evidence that UV exposure is associated with the prevalence of the majority of these genetic alterations. UV radiation also contributes to reactive oxygen species (ROS) accumulation and the maintenance of a prooxidant milieu, which, in turn, is associated with the activation of signaling pathways involved in melanomagenesis, including RAS/RAF/MEK/ERK and PI3K/AKT [12,13,14]. Therefore, melanoma can be considered a ROS-driven cancer, where different transduction pathways are activated to increase the survival, proliferation, and apoptosis resistance of melanoma cells.

Dysfunctional mitochondria and increased NADPH oxidase activity have been shown to be the major sources of ROS in melanoma, contributing to melanocyte malignant transformation and tumor progression [15,16,17]. Recently, our group and others showed that nitric oxide synthase (NOS) uncoupling is also involved in the redox homeostasis regulation of melanoma cells by increasing superoxide anion (O_2_^•^^−^) and decreasing nitric oxide (NO) production [18,19,20].

All three isoforms of NOS: endothelial (eNOS), neuronal (nNOS), and inducible (iNOS), require tetrahydrobiopterin (BH4) as a cofactor for NO synthesis. BH4 is indispensable for NOS enzymatic activity through mechanisms that include direct binding to the heme active site at the interface between the two monomers, promoting NOS dimerization and stability and increasing the binding of the substrate L-arginine [21,22]. Increased ROS concentrations in the intracellular milieu leads to the oxidation of BH4 to BH2, resulting in a BH4 loss. The decreased BH4 bioavailability contributes to NOS uncoupling, since electron transfer from NADPH through the flavin domains to molecular oxygen is uncoupled from L-arginine oxidation, resulting in the generation of ROS and NO production decrease [23,24].

BH4 is synthetized in physiological conditions mainly by a de novo pathway from GTP via three steps, the first and rate-limiting step catalyzed by GTP cyclohydrolase 1 (GTPCH1: EC 3.5.4.16) and, then, by the enzymes 6-Pyruvoyl-tetrahydropterin synthase (PTPS) and sepiapterin reductase (SPR) [25]. Intracellular BH4 concentration is also maintained by the salvage pathway, including SPR and dihydrofolate reductase (DHFR) enzymes, and by the recycling route where dihydropteridine reductase (DHPR) regenerates quinonoid dihydrobiopterin (qBH2) to BH4 [22,26].

BH4 metabolism alteration is involved in the development of pathologies associated with oxidative stress and inflammation, such as atherosclerosis, hypertension, diabetes, and neurological disorders, which is well-described [23,24,25,27]. However, BH4 contributions in cancer, where the redox disturbance represents a hallmark mechanism, is poorly understood. Analysis of the recent literature suggests that BH4 may have a dual role in tumor progression, which is dependent on the cancer type and disease stage development [20,28,29]. Accordingly, the goal of this work is to evaluate the participation of BH4 along melanoma progression by studying the cell lines corresponding to different stages of melanoma development.

## 2. Results

### 2.1. GCH1 Expression Is Elevated in Melanoma Cell Lines Compared to Melanocytes

Previous studies from our group showed that NOS is uncoupled in melanoma cells, and BH4 or L-sepiapterin (BH4 precursor) treatment increased NO production and reduced the O_2_^•−^ levels in these cells, suggesting NOS recoupling [19,20,30].

Since GTPCH1 is the rate-limiting enzyme in BH4 biosynthesis, the expression of the gene and the protein amount was evaluated in human cell lines corresponding to distinct stages of melanoma progression: melanocytes, WM1552C (RGP), WM793, and WM1366 (VGP), WM983B, and Lu1205 (MET). Initially, the expression of *GCH1* was analyzed by real-time qPCR and was found to be increased in WM1552C and WM983B melanoma cells compared to melanocytes. *GCH1* expression was higher in WM983B metastatic melanoma cells in relation to less aggressive melanoma cells WM1152C and WM793 (Figure 1A). This result was supported by GTPCH1 protein expression, where there was an increase in almost melanoma cell lines compared to melanocytes (Figure 1B,C).

### 2.2. Tetrahydrobiopterin Concentration Is Elevated in Metastatic Melanoma Cells

Our next step was to analyze the concentration of biopterins by HPLC. The BH4 amount was higher in WM983B metastatic melanoma cells when compared to melanocytes or even other melanoma cells. However, the BH4 concentration was lower in WM1552C RGP cells in comparison to melanocytes (Figure 2A). In addition, the BH4:BH2 ratio was also higher in WM983B metastatic cells compared to melanocytes and WM1552C RGP cells (Figure 2C). There was no difference in the BH2 concentration in the cell lines analyzed (Figure 2B).

### 2.3. Melanoma Cells Show Increased Superoxide Anion Levels and Decreased Nitric Oxide Concentration

We used the fluorescence indicator DHE and evaluate O_2_^•−^ levels by flow cytometry in melanocytes and melanoma cell lines. We found an increase in O_2_^•−^ in WM1552C, WM793, and WM983B melanoma cells when compared to melanocytes (Figure 3A) and intracellular nitric oxide amount by flow cytometry using the fluorescence indicator DAF. We found a decreased NO amount in WM1552C and WM983B melanoma cells compared to melanocytes (Figure 3B). We also analyzed the extracellular nitric oxide levels evaluated by the NO analyzer, and the results showed a decrease in the nitric oxide concentration in WM1552C and WM983B melanoma cells compared to melanocytes (Figure 3C).

### 2.4. Increased Expression of Nitric Oxide Synthases in Melanoma Cells

It has been demonstrated in endothelial and metastatic melanoma cells that increased the *NOS3* expression can cause eNOS uncoupling due to the loss of BH4:NOS stoichiometry [30,31,32]. Thus, we evaluated the expression of enzymes eNOS, iNOS, and nNOS in melanocytes and melanoma cells to investigate possible uncoupling mechanisms alongside melanoma progression, since we showed an augment in O_2_^•^^−^ and reduction of the NO levels in melanoma cells. We observed increased eNOS and nNOS expression in all melanoma cells compared to melanocytes (Figure 4A,B). iNOS expression was higher in WM793 melanoma cells when compared to other cells (Figure 4C). The results suggest that NOS uncoupling in WM793 and WM983B melanoma cells may be caused by the dysregulation in BH4:NOS stoichiometry. The same mechanism could also be proposed for WM1552C melanoma cells, which would also be associated with a lower concentration of BH4.

Additionally, melanoma cells were treated with specific NOS inhibitors to analyze their activity. In the presence of L-NAME, a preferentially eNOS inhibitor, the NO levels were decreased in WM1552C RGP (Appendix A) and WM983B metastatic melanoma cells (Appendix A), indicating that this isoform is a source of NO in these cells. Moreover, L-NAME increased the O_2_^•^^−^ amount in WM1552C (Appendix A) but had no effect on the O_2_^•^^−^ concentration of WM983B cells (Appendix A), suggesting that L-NAME can cause NOS uncoupling in WM1552C cells. On the other hand, NANT, a specific nNOS inhibitor, reduced the O_2_^•^^−^ levels in WM1552C (Appendix A) and WM983B melanoma cells (Appendix A). However, NANT increased the NO in WM1552C (Appendix A) and reduced it in WM983B melanoma cells (Appendix A), suggesting that only nNOS is uncoupled in these melanoma cells.

### 2.5. Tetrahydrobiopterin Supplementation Impairs Superoxide Anion and Increases Nitric Oxide Production in WM1552C Melanoma Cells

Showing a reduction in the BH4 amount and an increase in the expression of eNOS and nNOS compared to the melanocytes, we investigated whether the BH4 treatment would be able to restore enzyme coupling in the WM1552C RGP cells. Initially, the cells were treated with 20 or 40 µM of BH4, and after 16 h, the intracellular amount of biopterins was evaluated by HPLC. BH4 and the total biopterin concentration were higher in treated compared to untreated cells (Figure 5A,D). The treatment with 40 µM of BH4 increased the BH2 amount compared to untreated cells (Figure 5B), and we did not observe differences in the BH4:BH2 ratio (Figure 5C). Importantly, an increased BH4 concentration was maintained for 96 h (data not shown). The flow cytometry analysis showed that WM1552C melanoma cells decreased the NO and increased the O_2_^•−^ levels in the presence of BH4 (Figure 5E,F). These results indicate the ability of BH4 to restore the NOS function in the early phases of melanoma progression.

### 2.6. Tetrahydrobiopterin Inhibits the Growth of WM1552C Melanoma Cells

To evaluate the functional effects of NOS recoupling by BH4 treatment, with consequent changes in the redox homeostasis, we analyzed the WM1552C RGP viability by MTT treating cells with different concentrations of BH4. The treatment decreased the cell viability at 48, 72 and 96 h (Figure 6A). On the other hand, the treatment with BH4 did not modify the viability of the melanocytes [20]. Moreover, BH4 supplementation reduced the colony formation capability of WM1552C cells after 9 days (Figure 6B,C). Likewise, 40 µM of BH4 decreased the diameter in the forming tumorsphere when comparing the untreated group from the first day of the treatment (Appendix A) to the last day (day 5) (Figure 6D,E). In addition, BH4 increased the WM1552C melanoma cell sensitivity to vemurafenib (BRAF inhibitor) (Figure 6F).

### 2.7. DAHP Supplementation Potentiated ROS Production and NO Loss by Decreasing BH4 Levels in WM983B Metastatic Melanoma Cells

Since the WM983B metastatic melanoma cells showed increased BH4 levels and BH4:BH2 ratio when compared to melanocytes, in addition to a higher expression of eNOS and nNOS, we evaluated the effects of GTPCH1 inhibition on DAHP treatment. Analysis of the biopterin concentration by HPLC showed that the treatment with DAHP for 16 h decreased the BH4 amount, total biopterin and the BH4:BH2 ratio of the WM983B cells (Figure 7A,C,D). There was no difference in the BH2 levels (Figure 7B). Moreover, WM983B cells treated with 4mM DAHP showed increased O_2_^•−^ levels and decreased NO levels compared to the untreated cells (Figure 7E,F). On the other hand, treatment with BH4 increased the NO concentration (Appendix A) and reduced the O_2_^•−^ levels in WM983B cells (Appendix A), suggesting the recoupling of NOS in the presence of the NOS cofactor. However, BH4 did not interfere with the cell viability (Appendix A). These results suggest that the decrease in the BH4 amount potentiated NOS uncoupling in metastatic melanoma cells, while its increase restored NOS function.

### 2.8. DAHP Inhibits the Growth of WM983B Metastatic Melanoma Cells

Since the increase in NO and decrease in the O_2_^•−^ levels inhibited the growth of WM1552C RGP melanoma cells, we analyzed by MTT the viability of WM983C metastatic melanoma cells treated with DAHP. Surprisingly, 2 and 4 mM DAHP supplementation reduced the cell viability at 72 and 96 h (Figure 8A). In addition, DAHP treatment decreased the colony formation capability of WM983B cells after 9 days (Figure 8B). On the other hand, DAHP treatment at the same concentrations did not alter the viability of the melanocytes compared to the untreated cells (Appendix A).

### 2.9. Pertubance of BH4 Metabolism Impairs Oncogenic Signaling Pathways in Melanoma Cells

Since BH4 supplementation reduced WM1552C RGP melanoma cell proliferation, we analyzed the oncogenic signaling pathways involved in the growth and survival of melanoma cells. Surprisingly, ERK activation and AKT phosphorylation were induced in the presence of BH4 after 30 min and maintained for four hours in radial growth melanoma cells (Figure 9A). On the other hand, the reduction of WM983B cell growth after DAHP treatment seems to be associated with the inhibition of ERK and AKT phosphorylation (Figure 9B).

## 3. Discussion

Recent studies by our group showed that NOS uncoupling caused an increase in O_2_^•−^ and decreased NO levels in melanoma cells. A decreased BH4:BH2 ratio and enhanced NOS expression with consequent loss of NOS:BH4 stoichiometry have been indicated as a mechanism of an enzyme uncoupling in these tumor cells [20,30]. Furthermore, it has been suggested that uncoupled NOS is involved in the malignant transformation of murine melanocytes [12] and in the establishment of a prooxidant milieu, which, in turn, contributes to the proliferation and apoptosis resistance of human melanoma cells [17,27]. In the present work, we evaluated for the first time the participation of BH4 and NOSs activity status in human cell lines corresponding to distinct stages of melanoma progression.

As mentioned earlier, GTPCH1 is the limiting enzyme for BH4 synthesis via the de novo pathway, capable of converting GTP into 7,8-dihydroneopterin triphosphate. An increase in *GCH1*/GTPCH1 expression has been reported in several tumors [29,30,33,34]. Recently, through a murine melanoma model, we showed that there is an increase in the levels of *Gch1* and other genes involved in the de novo BH4 biosynthesis pathway during melanoma progression [30]. Accordingly, here, we observed that the *GCH1* gene levels and GTPCH1 protein expression are higher in WM1552C RGP and WM983B metastatic melanoma cell lines when compared to melanocytes (Figure 1A,B). Furthermore, WM983B cells had the highest concentration of intracellular BH4 when compared to other melanoma lines and melanocytes (Figure 2A). Surprisingly, WM1552C cells showed lower levels of BH4 when compared to melanocytes (Figure 2A), revealing that the increase in GTPCH1 expression was not enough to augment the BH4 concentration in these melanoma cells, as shown in endothelial cells [35]. Interaction with the enzyme GTP feedback regulator protein (GFRP) is one of the mechanisms that can regulate the activity of GTPCH1 in the endothelial, keratinocytes, and melanocytes [36,37]. It was demonstrated that, in the presence of high ROS levels, as observed in the early stages of melanoma progression, the relationship between GTPCH1 expression and BH4 amount is not linear, possibly due to the action of GFRP [38]. Since *GCHFR* expression is higher in WM1552C cells when compared to melanocytes (data not shown), a functional GFRP/GTPCHI axis could be the mechanism controlling the BH4 concentration in these melanoma cells, as found in melanocytes [37]. Furthermore, WM1552C cells showed no difference in BH2 concentration and BH4:BH2 ratio (Figure 2B,C). The concentration of BH2 is maintained by the salvage and recycling pathways, as was shown in endothelial and breast cancer cells [39,40,41]. Although WM1552C cells present a prooxidant status (Figure 3), the high expressions of *SPR, DHFR* and *DHPR* found in these melanoma cells (data not shown) can prevent BH4 oxidation and avoid a complete BH4 loss. Regarding WM983B metastatic melanoma cells, an increased GCH1/GTPCH1 expression is associated with a high BH4 concentration. A decreased expression of *GCHFR* in these metastatic cells could contribute to maintaining the BH4 concentration (data not shown). Importantly, in addition to the BH4 amount and BH4:BH2 ratio, the stoichiometric relationship between NOS and BH4 can determine the function of the enzyme [31,32,42]. The WM1552C RGP, WM793 VGP and WM983B metastatic melanoma cells showed higher eNOS and nNOS expressions when compared to the melanocytes (Figure 4A,B). Besides that, the iNOS expression was increased in the WM793 cells (Figure 4C). Moreover, all melanoma cell lines showed higher O_2_^•−^ levels and lower NO levels compared to the melanocytes. Thus, even in the presence of higher levels of BH4, the metastatic cell line WM983B seems to present NOS uncoupling, since the BH4 treatment of these cells was able to increase the NO levels and decrease O_2_^•−^ levels (Appendix A). Corroborating with this hypothesis, eNOS silencing in murine metastatic melanoma cells was able to increase the NO concentrations and decrease the O_2_^•−^ levels [30], indicating the enzyme recoupling by the reestablishment of BH4/NOS stoichiometry in different melanoma cells. On the other hand, we observed an increase in the O_2_^•−^ levels and a decrease in the NO concentration of WM983B cells treated with DAHP, a GTPCHI inhibitor (Figure 7E,F). The GTPCH1 inhibitor treatment reduced the intracellular amount of BH4 and the BH4:BH2 ratio (Figure 7A,B), potentiating NOS uncoupling. Furthermore, BH4 treatment was able to increase the intracellular levels of BH4 (Figure 5A), with the consequent reduction in O_2_^•−^ and increase in NO levels in WM1552C melanoma cells (Figure 5E,F), indicating the reestablishment of NOS function also in these cells.

Superoxide anion and other ROS are involved in different biological processes, acting on the oxidation of molecules and signal transduction like a second messenger. Under physiological conditions, there is a balance between the production and elimination of ROS [43]. Disruption of the redox homeostasis can promote several cancer types, including melanoma, which is in agreement with the literature [44,45]. ROS participate in the pathways related to proliferation, survival, invasion and metastasis [46,47,48,49]. On the other hand, ROS at high levels can have negative consequences, such as DNA damage and cell death, conferring an antitumor profile to these molecules. Many studies have reported ROS as a therapeutic strategy in the fight against cancer [44,50,51,52,53]. The treatment of molecules such disulfiram, brusatol and Buthionine sulfoximine render melanoma cells more sensitive to chemo- and radiotherapy through the modulation of redox homeostasis [54,55]. Interestingly, we found that DAHP treatment reduced the cell viability and clonogenic capacity of WM983B metastatic melanoma cells (Figure 8A,B), whereas the same treatment did not alter the melanocytes viability (Appendix A). These results suggest that the potentiation of NOS uncoupling induces oxidative stress, which, in turn, inhibits cell growth in the latest phases of melanoma progression, as observed by other authors [56,57]. On the other hand, decreased ROS and increased NO levels in the presence of BH4 did not impair the WM983B melanoma cells viability, indicating that, in this phase, low ROS did not interfere with melanoma development. Previous studies showed that NOS recoupling, through BH4 treatment or NOS silencing, was also able to decrease the cell viability, colony formation, *anoikis* resistance in melanoma cells and attenuated metastatic melanoma growth in vivo [20,30]. Here, we observed that BH4 treatment also reduced the cell viability, colony and tumorsphere formation of WM1552C RGP melanoma cells (Figure 6A–E). Increasing the ROS concentration is important in the early stages of cancer progression, as they over-activate oncogenic signaling pathways, as shown by other authors [12,18]. These data indicate that, either by recoupling or potentiating NOS uncoupling, changes in the superoxide anion and nitric oxide levels affected melanoma cell survival. It is important to note that the effect of redox status modulation in melanoma growth is dependent on the melanoma stage.

NO, which is also an important messenger molecule involved in different physiological processes, participates in tumorigenesis and tumor progression [58]. Generally, high concentrations of NO cause cell cycle arrest and apoptosis, while low concentrations stimulate proliferation [59,60]. In addition, the cancer cell type and localization of NO synthesis are also associated with its contribution to cancer. In melanoma, it has been shown that NO can also act to inhibit or promote tumor growth [58,61]. Among the mechanisms that NO can trigger cell death with is the activation of p53 [62,63,64]. Increase in iNOS and nNOS expression, and consequent higher NO production after treatment with capsaicin and resveratrol, stimulated p53 and promoted the apoptosis of human melanoma cells [65]. On the other hand, NO can act by activating the PI3K/AKT and RAS/ERK pathways, which, in turn, inhibit p53, favoring the development and progression of melanoma [66]. In fact, activation of these transduction pathways is associated with cancer development, including melanoma progression. However, in some circumstances, the induction of ERK and AKT activity can lead to cancer cell death [67,68,69]. As mentioned, ROS have a dual role in cancer development. It was shown that ROS activate innumerous signaling pathways, including the RAS/RAF/MEK/ERK and PI3K/AKT cascades, which, in turn, sustains melanoma growth [12,18,66]. However, the induction of ROS by different agents is associated with cell death in different cell types. Studies have shown that ERK activation by ROS can also lead to the death of different cell types [70,71,72,73]. For instance, Guon and colleagues found that the increase in ROS production caused the death of A2058 melanoma cells via ERK phosphorylation [73]. In human keratinocytes, TGF-α-induced ERK inhibition is caused by ionizing radiation/ROS [74]. Oxidative stress triggers the apoptosis of human endothelial cells by inhibiting ERK phosphorylation [75]. Here, we showed that DAHP treatment with the consequent reduction of NO and increase of the ROS levels was able to inhibit AKT and ERK phosphorylation in WM983B metastatic melanoma cells (Figure 9B). These results suggest that DAHP impaired the cell growth of WM983B cells by modulating the balance between NO and ROS. Surprisingly, BH4 treatment induced ERK activation and AKT phosphorylation in WM1552C RGP melanoma cells (Figure 9A), as shown by other authors [68,69,76]. Tangchirakhaphan showed that p-ERK1/2 triggers apoptosis and inhibits the cell growth of A375 melanoma cells treated with goniothalamin [76]. It was also demonstrated that serotonin type-3 receptor antagonists induce ERK activation and NF-κB downregulation-dependent apoptosis [69].

Interestingly, we observed a difference in the BH4 amount in the early and the advanced stage of melanoma. WM1552C RGP melanoma cells have a lower intracellular amount of BH4 when compared to melanocytes and WM983B, while the BH4 levels of the WM983B cells are higher than the nonmalignant and melanoma cells. Superoxide anion acts for malignant transformation, and low concentrations of BH4 could favor tumorigenesis. On the other hand, it is known that high concentrations of ROS can lead to cell death; thus, tumor cells can increase the antioxidant system to protect and ensure tumor progression [50,51,77]. In advanced-stage melanoma, the increase in BH4 levels may be one of the mechanisms to keep the O_2_^•−^ levels within the limit for the survival of these cells. Corroborating these arguments, despite showing differences in the BH4 amounts, we observed that the O_2_^•−^ levels do not change between the WM1552C and WM983B cells (Figure 3A).

The development of targeted therapies has brought greater perspective to patients with melanoma; however, the establishment of chemoresistance to these treatments has shown that new drugs need to be investigated. Several mechanisms of resistance development have been shown [5,6], such as the increase in ROS by the activation of oxidative phosphorylation [78,79]. Interestingly, we found that BH4 rendered WM1552C melanoma cells more sensitive than vemurafenib (Figure 6F). Furthermore, it was shown that vemurafenib triggers NO synthesis, being one of the mechanisms associated with vemurafenib-induced cell death [80]. Moreover, therapeutic strategies that potentiate NO production overcome the vemurafenib resistance [81]. Therefore, the mechanisms underlying the overcoming of vemurafenib resistance of WM1552C RGP melanoma cells may be the increment in the NO levels in the presence of BH4. BH4 acting as a NOS cofactor has been reported as a pro-tumoral and an antitumoral molecule [82]. Furthermore, since NOS can be a source of both ROS and NO, the altered BH4 concentration alongside the tumor progression with consequent modifications of cellular redox homeostasis can be explored as a therapeutic strategy according to the type and stage of the tumor.

## 4. Materials and Methods

### 4.1. Cell Culture

The human melanocyte cell line was acquired from Rio de Janeiro Cell Bank (code BCRJ 0190) and was grown in DMEMF12 (Invitrogen, Scotland, UK) at pH 7.2 supplemented with 20% fetal bovine serum (Invitrogen, Scotland, UK), 2.5 mM glutamine (Invitrogen, Scotland, UK), 500 μM sodium pyruvate (Invitrogen, Scotland, UK), 1.4 μM hydrocortisone (Sigma-Aldrich, St. Louis, MO, USA), 1 nM Triiodotreonin (T3) (Sigma-Aldrich, St. Louis, MO, USA), 10 μg/mL Insulin (Sigma-Aldrich, St. Louis, MO, USA), 10 μg/mL Transferrin (Sigma-Aldrich, St. Louis, MO, USA) and 10 ng/mL Epidermal Growth Factor (Sigma-Aldrich, St. Louis, MO, USA). The patient-derived melanoma cells lines WM1552C, WM793, WM1366, WM983B and Lu1205 were kindly provided by Dr. Meenhard Herlyn (Wistar Institute, Philadelphia, PA, USA) and grown in TU medium (80% MCDB153 (Sigma-Aldrich, St. Louis, MO, USA medium plus 20% Leibovitz medium (Invitrogen, Scotland, UK) supplemented with 2% FBS and 1.68 mM CaCl_2_ (https://wistar.org/our-scientists/meenhard-herlyn, accessed on 1 April 2022). WM155C lineage corresponds to radial growth phase (RGP) melanoma cells, WM793 and WM1366 lineages correspond to vertical growth (VGP) melanoma cells and WM983B and Lu1250 correspond to the metastatic melanoma cells. The cultures were kept in an incubator at 37 °C in a humidified atmosphere containing 5% CO_2_.

### 4.2. Nitric Oxide Quantification

#### 4.2.1. DAF-2DA

Cells were grown in 6-well plates and, upon reaching 85% confluence, were treated or not for 16 h with 10 and 40 μM BH4 (Cayman Chemical, Ann Arbor, MI, USA) or 2 and 4 mM DAHP (2,4-Diamino-6-hydroxypyrimidine) (Cayman Chemical, Ann Arbor, MI, USA) or for 45 min with 0.5, 1 and 2 mM L-NAME (Nitro-L-arginine methyl ester hydrochloride (Sigma-Aldrich, St. Louis, MO, USA) and 100, 200 and 400 nM NANT (N-[(4S)-4-amino-5-[(2-aminoethyl)amino]pentyl]-N’-nitroguanidinetris (trifluoroacetate) (Cayman Chemical, Ann Arbor, MI, USA). After 16 h of incubation, cells were washed with phosphate-buffered saline (PBS) and incubated with 2.5 μM 4-5-diaminofluorescein diacetate (DAF-2DA), a nonfluorescent cell permeable indicator for nitric oxide, in PBS (Molecular Probes, Eugene, OR, USA) for 40 min at 37 °C in the dark. Cells were washed two times with PBS and analyzed by flow cytometry (FACSCalibur; Becton–Dickinson, Franklin Lakes, NJ, USA) (excitation wavelength 495 nm; emission wavelength 515 nm).

#### 4.2.2. NO Analyzer

NO concentration was determined after a gas-phase chemiluminescence reaction of NO with ozone by a NO analyzer (NOA 280; Sievers Instruments, Boulder, CO, USA). A standard curve was established with a set of serial dilutions (0.1–100 μM) of sodium nitrate. The amount of NO metabolites in melanocytes and in WM1552, WM793 and WM983 melanoma cells were determined by comparison with the standard curve and expressed as micromoles per milligram of protein. Data collection and evaluation were performed using the NOAnalysis software (version 3.21; Sievers Instruments, Boulder, CO, USA).

### 4.3. Reactive Oxygen Species Anion Quantification

Intracellular reactive oxygen species amount was measured using CellROX^®^ Green (Molecular Probes, Eugene, OR, USA), a nonfluorescent cell permeable indicator and analyzed by flow cytometry. After 16 h of treatment or not with 20 and 40 μM BH4 or 2 and 4 mM DAHP) or for 45 min with 0.5, 1 and 2 mM L-NAME and 100, 200 and 400 nM NANT cells were washed and incubated with 1 μM CellROX in PBS for 50 min at 37 °C in the dark. After washing, cells were analyzed by flow cytometry (FACSCalibur; Becton–Dickinson, Franklin Lakes, NJ, USA) (excitation wavelength 480 nm; emission wavelength 567 nm).

### 4.4. High-Performance Liquid Chromatography Analysis of the Cellular Biopterin Content

To evaluate the cellular concentrations of tetrahydrobiopterin (BH4), 7,8-dihydrobipterin (BH2), total biopterin and the BH4:BH2 ratio, reversed-phase high-performance liquid chromatography was used as a previously reported [83]. Melanoma cell lines and treated cells with 20 and 40 μM BH4 or 2 and 4 mM DAHP for 16 and 96 h were washed twice with cold PBS (10 mM phosphate buffer, pH 7.2, 150 mM NaCl). After centrifugation, cells were resuspended in 0.5 mL 0.1M phosphoric acid containing 5 mM dithioerythritol and sonicated for 40 s, to which 35 µL 2 M trichloroacetic acid (TCA) were added. The solution was centrifuged at 12,000× *g* for 1 min, and the supernatant was used immediately for the quantification of all biopterins. The total biopterin amount was measured following oxidation in acidic conditions, whereas BH2 quantification was conducted after its oxidation in alkaline conditions. BH4 was calculated from the difference between the amount of biopterin formed by oxidation in acidic conditions and the amount of biopterin formed by oxidation in alkaline conditions. For an oxidation reaction under acidic conditions, 100 µL of cell extract were mixed with 15 µL 0.2 M TCA and 15 µL 1% I_2_/2% KI in 0.2 M TCA. For oxidation under alkaline conditions, 100 µL cell extracts were mixed with 15 µL 1 M NaOH and 15 µL 1% I_2_/2% KI in 3 M NaOH. The oxidation reaction was carried out for 1 h in the dark at room temperature. The next step was to inactivate the excess of iodine by the addition of 25 µL 0.114M ascorbic acid. The assay mixture was centrifuged at 4 °C for 12 min, and 100 µL of the supernatant were injected into an HPLC system (LCMS-2020, Shimadzu Co., Kyoto, Japan) on a C18 Vydac reversed-phase column (5 μm, 4.6 mm id × 205) and detected by fluorescence (lex = 350 nm; lem = 450 nm). Biopterin was eluted by an isocratic mobile phase solution (5% methanol and 7.5 mM sodium phosphate buffer, pH 6.35) at a flow rate of 1.0 mL/min. Data were collected and analyzed by LC solution software (Shimadzu Co., Kyoto, Japan) and normalized against protein concentration.

### 4.5. RNAm Analysis

RNA was isolated using Trizol (Invitrogen, Carlsbad, CA, USA), according to the manufacturer’s specifications. cDNA was synthetized using the Superscript III first-strand synthesis system following the manufacturer’s instructions (Invitrogen, Carlsbad, CA, USA). Equal amounts of each cDNA were quantified by real time-PCR in a Corbett Rotor-Gene 6000 Detection System version 1.7 using an SYBR green PCR master mix (Qiagen, Dusseldorf, German) and specific primers (*GCHI* forward: 5′ TGAGATGGTGATTGTGAAGGAC 3′; *GCHI* reverse: 5′ CGCTCCTGAACTTGTAGTCTTC 3′ Relative quantification (RQ) of the amplicons was calculated according to the 2^−∆∆Cq^ method. Gene normalizing was performed using β-actin (Forward: 5′ GTCTTCCCCTCCATCGTG3′; Reverse: 5′ GTACTTCAGGGTGAGGATGC 3′.

### 4.6. Western Blot

After treatment with 20 and 40 μM BH4 or 2 and 4 mM DAHP for 30 min and 4 h, subconfluent cell cultures were trypsinized, washed twice with PBS and whole-cell lysates were prepared using Pierce^®^ IP Lysis Buffer (25 mM Tris•HCl pH 7.4, 150 mM NaCl, 1% NP-40, 1 mM EDTA and 5% glycerol) added with the protease and phosphatase inhibitor cocktails (Sigma-Aldrich, St. Louis, MO, USA) kept for 15 min on ice, followed by centrifugation at 13,000 rpm for 15 min at 4 °C. The supernatant was collected, and the protein concentration was measured by Bio-Rad protein assay dye reagent concentrate (Bio-Rad, Hercules, CA, USA). Equivalent amounts of protein (40 μg) were denaturated in SDS sample buffer (240 mM Tris–HCl, pH 6.8, 0.8% SDS, 200 mM beta-mercaptoethanol, 40% glycerol and 0.02% bromophenol blue) for 5 min. Protein lysates were resolved by SDS-PAGE and transferred onto nitrocellulose membranes (Bio-Rad, Hercules, CA, USA). After protein transfer, the membranes were blocked with 5% nonfat dry milk in PBS. The specific antibody used was rabbit anti-GTPCH1 (Abnova, Taipei, Taiwan), rabbit anti-pan AKT, rabbit anti-phospho-AKT, rabbit anti-total ERK and rabbit anti-phospho-ERK (Cell Signaling Technologies, Danver, MA, USA), followed by secondary antibody incubations using goat anti-rabbit IgG (H + L) HPR-conjugated (1:2000) (Bio-Rad Hercules, CA, USA). The signal was visualized by chemiluminescence using Immobilon Forte Western HRP substrate (Merck KGaA, Darmstadt, Germany). The band intensities were measured using Processing and Analysis in Java, ImageJ 1.38b (Wayne Rasband, National Institute of Health, USA, http://rsb.info.nih.gov/ij/, accessed on 3 April 2021).

### 4.7. MTT Assay

Cell viability in the presence of BH4, DAHP or PLX4032 was determined using a standard MTT assay. After treatment or not with 10, 20 and 40 μM BH4 or 1, 2 and 4 mM DAHP or PLX4032 (0,5, 1, 2, 4, 8, 12, 16 and 20 μM) for 24, 48, 72 and 96 h, the human melanocytes and melanoma cells were submitted to the MTT assay. BH4 treatment was replaced every 24 h and cells were growing at 37 °C in a humidified atmosphere containing 5% CO_2_. As soon as the cells adhered to the substrate, 5 mg/mL MTT (Calbiochem, Hesse, Germany) was added to the culture, being considered as time T = 0. Cells were kept in an incubator at 37 °C and 5% CO_2_, with MTT for one hour. After medium withdrawal, dimethylsulfoxide (Merck, Hesse, Germany) was added to all wells for 15 min, and the absorbance was measured on a spectrophotometer at 620 nm (Multiskan EX, Thermo Electron, Stockport, OH, USA).

### 4.8. Clonogenic Assay

Melanoma cells were seeded on 60-mm dishes and grown for nine days in the presence or not of 40 μM BH4 or 2 and 4 mM DAHP. The cell culture medium was changed daily, and the same treatment as that of the initial plating was used. At the end of this period, the plates were washed in PBS, fixed in 3.7% (*v*/*v*) formaldehyde for 15 min, washed with PBS, stained with 1% Toluidine Blue in 1% sodium tetraborate (borax) for five minutes and washed with water. For the quantification of the surviving cells, the dye was solubilized in 1% SDS under agitation for one hour, and the absorbance at 620 nm was measured using a spectrophotometer.

### 4.9. Tumorsphere-Forming Assay

To evaluate the 3D cell proliferation, a spheroid formation assay was performed in micro-molds containing 81 pores. Approximately 700 µL of the 2% agarose solution in saline was added to the micro-mold (MicroTissues^®^ 3D Petri Dish^®^ micro-mold spheroids; Sigma, St. Louis, MO, USA). After 30 min, the micro-molded agarose was misinformed from the micro-mold in a 24-well plate. After the agarose was completely dried, approximately 20 min, DMEM containing 2.5% SFB was added to micro-molded agarose and was incubated for 15 min in an incubator at 37 °C and 5% CO_2_. After removing the medium, the cell suspension (approximately 1 × 10^6^ cells) was resuspended in 200 µL of medium and added to the micro-molded agarose. After spheroid formation (24 h later), cells were treated or not with 20 and 40 μM BH4 or 2 and 4 mM DAPH for 5 days, and sphere growth was monitored for 5 days, and the tumorsphere area was measured.

### 4.10. Statistical Analysis

To evaluate the results, we relied on the Student’s *t*-test for two-group experiments and analysis of variance (factorial ANOVA) with Bonferroni’s post-test for experiments with three or more groups using the GraphPad Prism 7.0^®^ statistical software (GraphPad, San Diego, CA, USA). The significance level was established at *p* < 0.05.

## Figures and Tables

**Figure 1 ijms-23-05979-f001:**
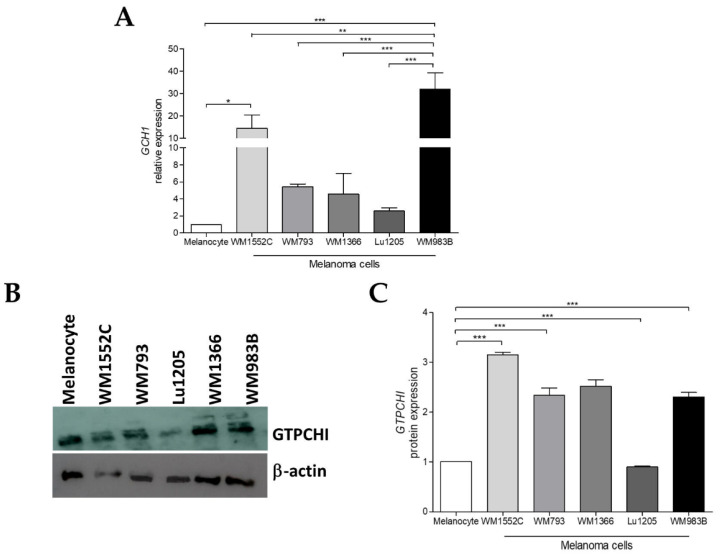
*GCH1* expression is higher in melanoma cells. (**A**) Relative mRNA levels of *GCH1* were determined in melanocytes, WM1552C, WM793, WM1366, Lu1205, and WM983B melanoma cells by real-time qPCR. (**B**,**C**) Protein expression of GTPCH1 in melanocytes and melanoma cells were determined by Western blot using a specific antibody. β-actin was used as the internal control. The Western blot image shows a representative result of three independent experiments. Values are reported in the bar graphs and expressed as the means ± S.D. The experiments were performed in triplicate, and *p*-values were based on the one-way ANOVA test, followed by Bonferroni’s post-test; *, *p* < 0.05; **, *p* < 0.01; and ***, *p* < 0.001.

**Figure 2 ijms-23-05979-f002:**
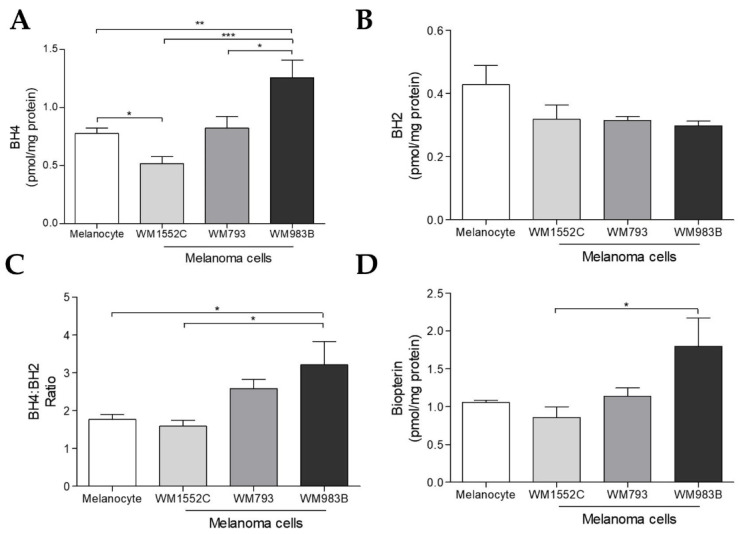
Tetrahydrobiopterin concentration is higher in metastatic melanoma cells. The amounts of BH4 (**A**); BH2 (**B**), BH4:BH2 ratio (**C**), and total biopterin (**D**) of melanocytes, WM1552C, WM793, and WM983C melanoma cells were determined by HPLC. Values are reported in the bar graphs and expressed as the means ± S.D. The experiments were performed in quintuplicate, and *p*-values were based on the one-way ANOVA test, followed by Bonferroni’s post-test; *, *p* < 0.05; **, *p* < 0.01; and ***, *p* < 0.001.

**Figure 3 ijms-23-05979-f003:**
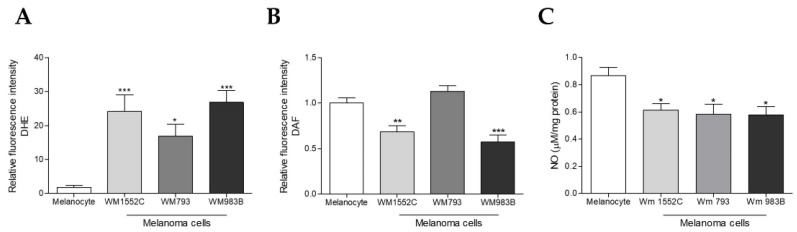
Melanoma cells show increased superoxide anion levels and decreased nitric oxide concentrations. (**A**) O_2_^•−^ amount in melanocytes and in WM1552C, WM973, and WM983B melanoma cells was analyzed by flow cytometry using DHE. The NO amount was evaluated by flow cytometry using DAF (**B**) or by the NO analyzer (**C**). Values are reported in the bar graphs and expressed as the means ± S.D. The experiments were performed in triplicate, and *p*-values were based on the one-way ANOVA test, followed by Bonferroni’s post-test; *, *p* < 0.05; **, *p* < 0.01; and ***, *p* < 0.001.

**Figure 4 ijms-23-05979-f004:**
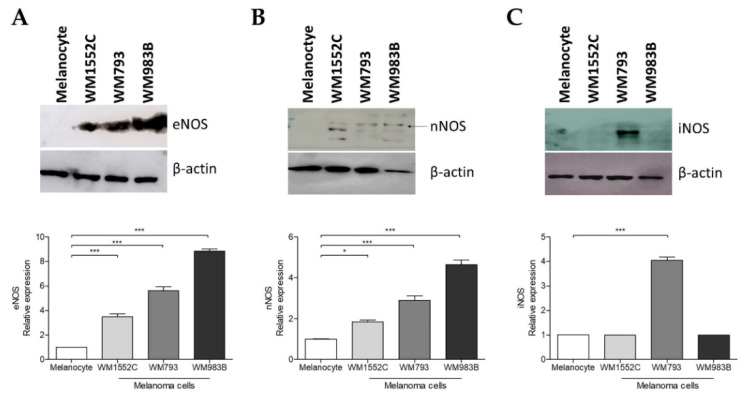
Melanoma cells exhibit increased protein expression of nitric oxide synthases. Expressions of eNOS (**A**), nNOS (**B**), and iNOS (**C**) in melanocytes and in WM1152C, WM793, and WM983B melanoma cells were determined by Western blot using specific antibodies. The Western blot images show representative results of three independent experiments. Values are reported in the bar graphs and expressed as the means ± S.D. The experiments were performed in quintuplicate, and *p*-values were based on the one-way ANOVA test, followed by Bonferroni’s post-test; *, *p* < 0.05 and ***, *p* < 0.001.

**Figure 5 ijms-23-05979-f005:**
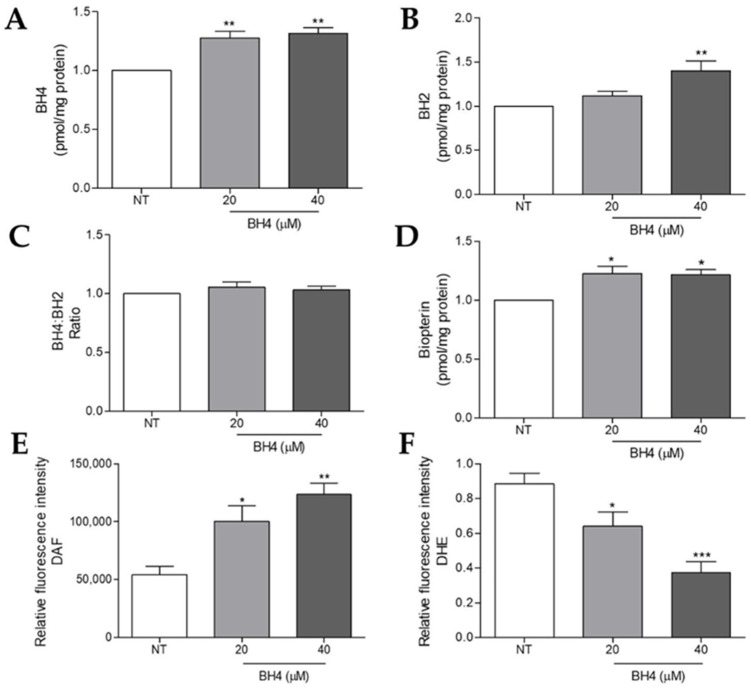
Tetrahydrobiopterin treatment restored the intracellular BH4 and recoupled NOS activity in radial growth melanoma cells. WM1552C cells were treated or not for 16 h with 20 and 40 μM BH4 and the amounts of BH4 (**A**), BH2 (**B**), BH4:BH2 ratio (**C**), and total biopterin (**D**) were determined by HPLC. The NO amount was evaluated by flow cytometry using DAF (**E**) and the O_2_^•^^−^ levels using DHE (**F**). Values are reported in the bar graphs and expressed as the means ± S.D. The experiments were performed in triplicate, and the *p*-values were based on the one-way ANOVA test, followed by Bonferroni’s post-test. * *p* < 0.05, ** *p* < 0.01 and *** *p* < 0.001.

**Figure 6 ijms-23-05979-f006:**
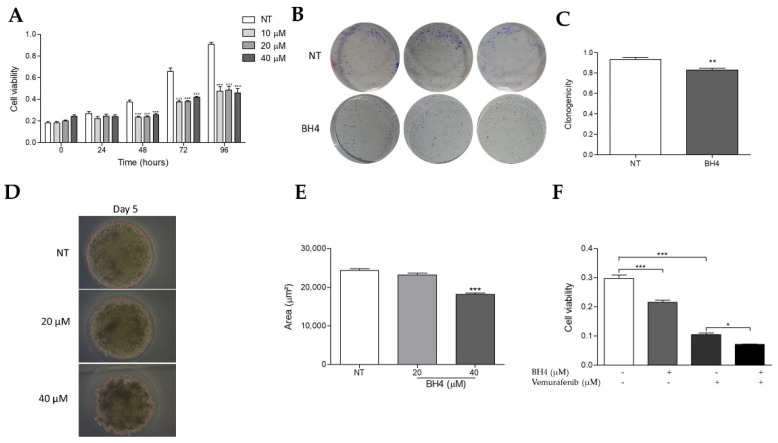
Tetrahydrobiopterin impaired the radial melanoma cell growth. WM1552C melanoma cells were treated (BH4) or not (NT) for 24, 48, 72 and 96 h with 10, 20 and 40 μM BH4, and viable cells were evaluated by MTT (**A**) for 9 days with 40 μM BH4, and the formation of clones was visualized by the clonogenicity assay (**B**,**C**) or for 5 days with 20 and 40 μM BH4, and the formation of tumorspheres was evaluated by the bead diameter measure (**D**,**E**). The viability of melanoma cells was also evaluated in the presence of 40 μM BH4 and 8.6 μM vemurafenib (**F**). Values are reported in the bar graphs and expressed as the means ± S.D. The experiments were performed in triplicate. *p*-values were based on the one-way ANOVA test, followed by Bonferroni’s post-test and two-way ANOVA test, followed by Bonferroni’s multiple comparisons test or by the Students’ *t*-test. *, *p* < 0.05; **, *p* < 0.01 and ***, *p* < 0.001.

**Figure 7 ijms-23-05979-f007:**
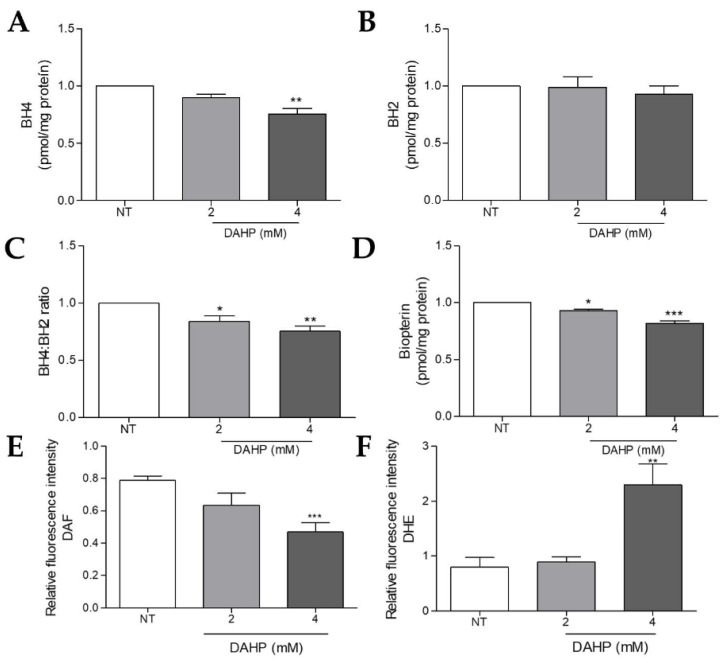
DAHP treatment reduced the intracellular BH4 and intensified the NOS uncoupling. WM983B cells were treated or not for 16 h with 2 and 4 mM DAHP, and the amounts of BH4 (**A**), BH2 (**B**), BH4:BH2 ratio (**C**), and total biopterin (**D**) were determined by HPLC. The NO amount was evaluated by flow cytometry using DAF (**E**) and the O_2_^•−^ levels using DHE (**F**). Values are reported in the bar graphs and expressed as the means ± S.D. The experiments were performed in triplicate, and the *p*-values were based on the one-way ANOVA test, followed by Bonferroni’s post-test *, *p* < 0.05; **, *p* < 0.01 and ***, *p* < 0.001.

**Figure 8 ijms-23-05979-f008:**
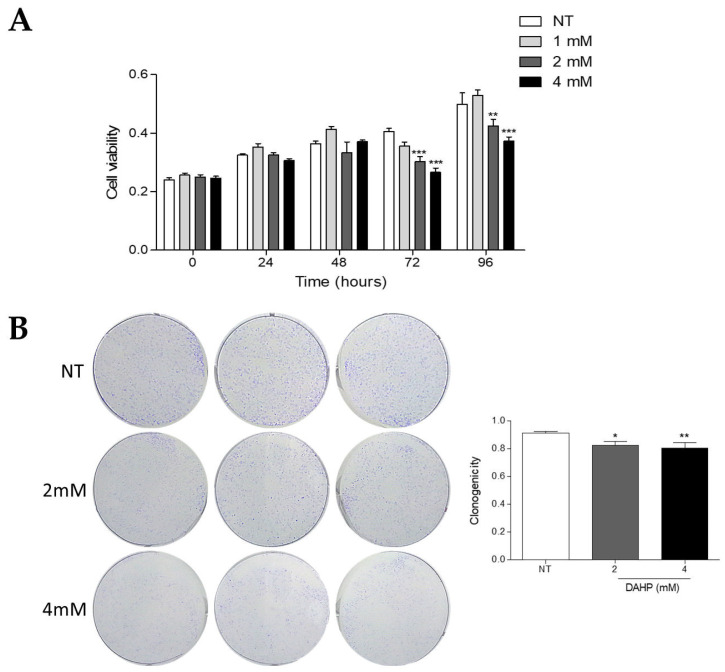
DAHP reduced the metastatic melanoma cell growth. WM983B melanoma cells were treated (DAHP) or not (NT) for 24, 48, 72 and 96 h with 1, 2 and 4 mM DAHP, and viable cells were evaluated by MTT (**A**) and for 9 days with 2 and 4 mM DAHP, and the formation of clones was visualized by a clonogenicity assay (**B**). Values are reported in the bar graphs and expressed as the means ± S.D. The experiments were performed in triplicate. *p*-values were based on a one-way ANOVA test, followed by Bonferroni’s post-test and two-way ANOVA test, followed by Bonferroni’s multiple comparisons test. *, *p* < 0.05; **, *p* < 0.01 and ***, *p* < 0.001.

**Figure 9 ijms-23-05979-f009:**
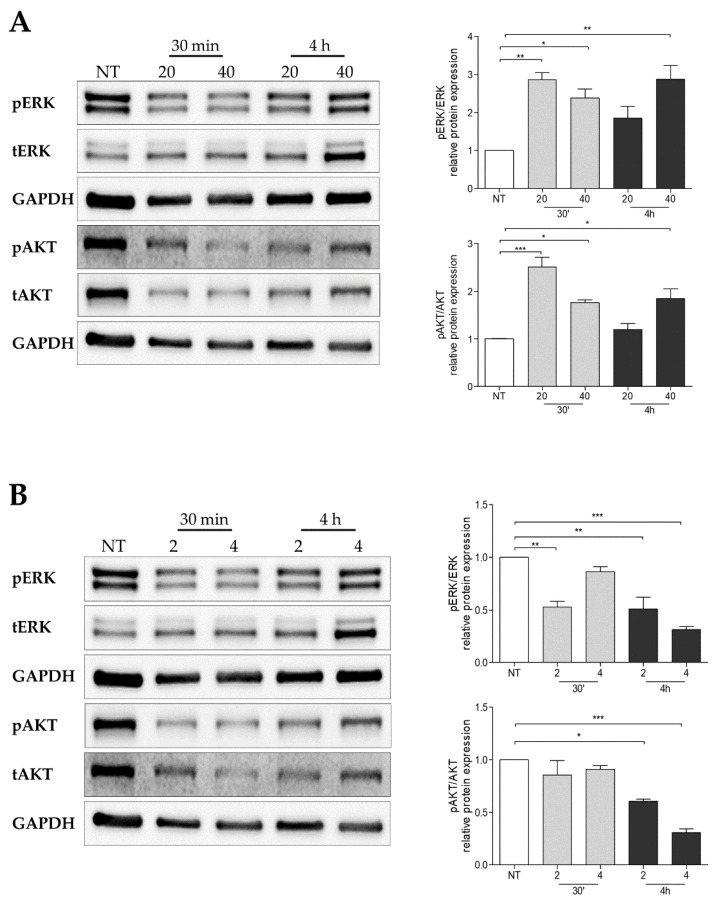
Alterations of BH4 amounts that altered the ERK and AKT oncogenic signaling pathways. Activation of the ERK and AKT pathways was analyzed in WM1552 radial growth melanoma cells treated with 20 and 40 μM BH4 for 30 min and 4 h (**A**) and, in WM983B metastatic melanoma cells, treated with 2 and 4 mM DAHP for 30 min and 4 h (**B**) by Western blot using specific antibodies for phospho- and total ERK and phosphor- and total AKT. GAPDH was used as the internal control. The Western blot images show the representative results of three independent experiments. Values are reported in the bar graphs and expressed as the means ± S.D. The experiments were performed in triplicate, and the *p*-values were based on the one-way ANOVA test, followed by Bonferroni’s post-test; *, *p* < 0.05, **, *p* < 0.01 and ***, *p* < 0.001.

## Data Availability

Not applicable.

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
