# Peer review of "Disruption of Redox Homeostasis by Alterations in Nitric Oxide Synthase Activity and Tetrahydrobiopterin along with Melanoma Progression"

_ijms, 2022, doi:10.3390/ijms23115979_

Round 1

Reviewer 1 Report

General comment :

Along the manuscript, BCRJ0190 cells are expected to represent a representive melanocyte line ; however, due the age of the donnor patient (1 year old ; BCRJ0190), the comparision with melanoma cell lines (age of resection, somatic origin, PDs at time of experiments, etc.) is questionable. In general, the manuscipt is difficult to follow and not all data may support the conclusions drawn.

Figure 1. I don’t understand the organization of the mount. e.g. it is quite unusual to present WB following quantitation. Further the quality of the western blot presented is quite poor. Since the C panel (which should be D) is supposed to illustrate a representative experiment (out of at least 3 independent experiments). I also wonder why panel B is included in this mount. It would probably be better to present levels of GCH1 together with Kaplan-Meyer’s graphs before results corrrelation in human melanoma cell lines and the stage at which they where isolated.

Figure 2.

I wonder why panel B does not include statistics ?

Figure 3.

There is some discrepancy between panel B and C, (WM793). How may the authors explain or at least discuss this.

Figure 4.

The quality of WBs is quite poor. Under these circumsatnces I have some concerns about quantitative approaches.

Figure 5 .

All results are obtained from WM1552C cells. Why ?

Figure 6.

Again, only WM1552C cells are used in this series of experiments. In the pictures I have, the clonogenic assays are poorly convincing. Idem with the organoid experiments. Perhaps, better pictures would deserve better quantitation understanting. Also, for a better presentation, organoids diameters could be highlighted by dashed circles.

Figure 7. WM983B cells, the reason why these cells were chosen here is unclear. This adds more difficulties to follow the manuscript.

Figure 8. WM983B, again this a difficulty to follow the manuscript. Results claimed based on this figure (notably B) are not obvious observing the microphotographs I have.

Figure 9. Comparision between WM1552 versus WM983B is unclear to me. Further, these experiments are not quantified and not subjected to statistical analyses.

In summary, conclusions of the manuscript should be supported by the use of more melanocytes and melanoma cell lines cultured under the same medium conditions.

To me, under the present methodology approaches and results presentation, conclusions are quite premature.

Reviewer 2 Report

The manuscript “Disruption of redox homeostasis by alterations in nitric oxide synthase activity and tetrahydrobiopterin along with melanoma progression” is a research article regarding the possibility that disruption of cellular redox homeostasis by altered BH4 concentration could be used as a therapeutic strategy for melanoma treatment, depending on the stage of the disease. I appreciate this manuscript that is well written and easy to read, the conclusions are supported by results and the work in general will be interesting for the journal readers. However there are some concerns that authors should address before the manuscript could be considered for publication:

  1. Figure 1: why was GCH1 expression valuated only at mRNA level, while for GTPCH1 authors performed western blot analysis?
  2. In the introduction section authors should provide more details about melanoma impact; indeed, authors correctly state that cutaneous melanoma is the most aggressive type of skin cancer, but they should also underline that cutaneous malignant melanoma represents less than 5% of all malignant skin neoplasms. Furthermore, authors should provide more details about the classification system of melanoma, which differs from other solid tumors (PMID: 34638427).
  3. Line 419: “was grown in DMEMF12 at pH 7.2 supplemented with and supplemented..” please correct.
  4. Line 420: “1,4 uM” should be written “1.4”. Several mistakes of this type are present in material and methods. Please correct replacing comma with a dot everywhere through the text (e.g. line 433 “0,5” etc…)
  5. Please provide details about the manufacturer of serum and media.
  6. Line 443: a dot in “NO analyzer.” should be removed.
  7. Line 486: “RNAm Expression Analysis” : do authors mean mRNA ? Please correct.
  8. Line 488: Please provide details about cDNA synthesis and primers used for the Real-Time PCR.
  9. Line 491: “2-∆∆Cq method”: -∆∆Cq must be written in upperscript.
  10. Line 507: please provide details about blocking and manufacturer and dilution of the secondary antibody.
  11. In the discussion section (line 335) authors underline that all melanoma cell lines showed higher O2•- levels and lower NO levels compared to melanocytes. This is a very important result and authors should underline that these results are in accordance with what is reported in literature, since in comparison with other solid tumors, ROS are particularly elevated in melanomas (PMID: 35453297). Indeed, increasing ROS production seems to be a promising therapeutic strategy for enhancing chemotherapy efficiency in melanomas (PMID: 33297311). Please integrate the paragraph with these considerations.

Author Response

Please see the attachament. 

Round 2

Reviewer 1 Report

The authors have revised the manuscript adequately.